# Body Image and Psychological Impact of Dental Appearance in Adolescents with Malocclusion: A Preliminary Exploratory Study

**DOI:** 10.3390/children10101691

**Published:** 2023-10-16

**Authors:** Federica Sicari, Emanuele Maria Merlo, Giulia Gentile, Riccardo Nucera, Marco Portelli, Salvatore Settineri, Liam Alexander MacKenzie Myles, Angela Militi

**Affiliations:** 1Department of Cognitive Sciences, Psychology, Educational and Cultural Studies (COSPECS), University of Messina, 98100 Messina, Italy; federica.sicari@unime.it; 2Department of Biomedical, Dental Science and Morphological and Functional Images, University of Messina, 98100 Messina, Italy; giulia97gentile@libero.it (G.G.); riccardo.nucera@unime.it (R.N.); marco.portelli@unime.it (M.P.); settineri@unime.it (S.S.); angela.militi@unime.it (A.M.); 3Department of Experimental Psychology, University of Oxford, Oxford OX2 6GG, UK; liam.myles@stx.ox.ac.uk

**Keywords:** body image, clinical psychology, malocclusion, oral health, dental treatment innovation

## Abstract

Background: Body image and psychosocial functioning represent central challenges during adolescence and early adulthood. Malocclusion, defined as an irregularity in the alignment of the teeth, is known to negatively influence psychological outcomes. The current study aimed to elucidate the role of malocclusion, together with age, gender, and dental class, in body image and psychological functioning. Methods: A total of 126 participants aged from 12 to 19 years old (mean: 15.87, SD: 2.35, female participants: 52.4%, male participants: 47.6%) were recruited. Participants were visited at the University Hospital of Messina, Italy, and completed a sociodemographic questionnaire, the Body Image Concern Inventory (I-BICI), and the Psychosocial Impact of Dental Aesthetics Questionnaire (PIDAQ). Results: Significant correlations were found between age, dental class, the BICI, and the PIDAQ. In particular, age showed a positive and significant correlation with PIDAQ—total score. The correlations between occlusal status and the BICI variables were all significant and positive. All correlations between occlusal status and the PIDAQ variables were all significant and positive, except for dental self-confidence. The correlations between the variables of the PIDAQ and BICI instruments were all significant and positive, except for dental self-confidence, where the directions were significant and negative. Moreover, age, gender, and occlusal status predicted BICI and PIDAQ scores. Age was a positive predictor for PIDAQ self-confidence, gender for BICI and PIDAQ total scores, along with dysmorphic symptoms, social impact, psychological impact, and aesthetic concerns. Several significant gender differences were highlighted by the analyses, with higher scores in the female group on all the BICI variables, except symptom interference, and all the PIDAQ variables, except dental self-confidence. Conclusions: Malocclusion appeared to play a central role in the psychological, representational, and psychosocial life of the participants. This research suggests that malocclusion and dental issues influence the psychological, representational, and psychosocial life of adolescents. Further research is required to examine the psychological impact of dental problems.

## 1. Introduction

Malocclusion is defined as an irregularity in the alignment of the teeth or their relationships during dental occlusion beyond the norm [1]. This involves the two dental arches approaching each other when the jaws close. Dental problems deriving from malocclusion often negatively affect psychological functioning, particularly with respect to body image. 

Body image has a central influence on psychological welfare during adolescence, a period of profound transformations in bodily representations [2]. This phase is characterized by somatic changes during pubertal development and changes in both one’s personality and psychological flexibility [3,4]. As discussed in the scientific literature, the presence of psychopathology during childhood, adolescence, and early adulthood constitutes a contentious matter [4,5,6]. Some research suggests that adolescents evaluate their image on the basis of appearance, such as weight and facial appearance [7]. Relatedly, one’s mouth entails an important role in body image, and influences biopsychosocial functioning [8]. Teeth acquire an aesthetic value with sexual and relational implications, such that imperfections (a societally-determined construct) can influence self-esteem and social acceptance [9]. The value assumed by the teeth during adolescence can impact craniofacial anomalies, which can contribute to psychological difficulties [10]. 

Several studies have analyzed the impact of dentofacial appearance on adolescent body perception [11,12], and have highlighted that alterations, such as malocclusions [13,14,15], and negative body image impact in terms of controlling behaviors, dissatisfaction, and mental health concerns [16,17,18,19].

Malocclusion is included by the World Health Organization (WHO) [20] in the category “Handicapping Dento Facial Anomaly” and is defined as "an alteration of craniofacial anatomy capable of influencing functionality, facial structure and psychological well-being”. According to Houston et al. [21], malocclusion refers to a situation where teeth of the two arches do not align perfectly; this involves an imbalance in the size and position of facial bones and soft tissues, such as the lips, tongue, and cheeks. Edward Angle [22] classified malocclusion into first, second, and third classes based on the alignment of the teeth. According to the angle and as reported by Proffit and colleagues [23], classes concern an incorrect line during occlusion due to misalignment, rotations, or other issues (Class I), lower molars in a distal position against upper equivalent with no reference to the given line of occlusion (Class II), and lower molar in a mesial position with reference to its counterpart with no reference to the given line of occlusion (Class III).

These conditions entail not only implications for one’s appearance but also functional effects, such as difficulty in swallowing and pronunciation; this can cause psychophysical discomfort [24]. Some types of malocclusions, if associated with severe dental crowding, can worsen hygiene, increasing the risk of cavities, gingivitis, or periodontal disease, and potentially further undermine the patient’s body image and social relationships [25].

Cunningham and O’Brien [26] suggested that the most significant impact of malocclusion is expressed in the psychosocial domain. Occlusive deficits, associated in particular with the second and third classes, alter the facial biotype and may contribute to the onset of psychological difficulties [27,28].

Age and gender also appear to play a fundamental role in the consideration of appearance and psychological difficulties deriving from malocclusion. The acquisition of body awareness in subjects aged between 12 and 15 has a significant impact of dental appearance, particularly on self-esteem and psychosocial functioning [29,30,31,32,33]. Also, research suggests that female adolescents give more aesthetic value to dental characteristics than males [34,35], perhaps because society regards the face and smile as significant aspects of beauty and there are significant societal pressures for women to abide by beauty standards [36]. According to Johnston et al. [37], the number of female subjects who request an orthodontic consultation is higher than males. In fact, dental appearance has a greater impact on quality of life on women, as compared to men [38].

It is necessary to explore the psychological implications of malocclusion to aid understanding of potential interventions to improve wellbeing. Any treatments should improve oral-health-related quality of life (OHRQoL), physical health, the level of self-esteem, and psychological wellbeing [39]. Indeed, oral health has also been reported to influence one’s bodily satisfaction [40,41,42]. The aim of the present study was to elucidate whether sociodemographic variables, psychological variables, and psychosocial variables are significantly related. The scope of the analyses was to elucidate any relationships between the abovementioned variables through examining correlations and gender differences. Accordingly, the following hypotheses were provided.

### Study Hypotheses

Firstly, it was hypothesized that there would be significant correlations among age, occlusal status, psychosocial impact of dental aesthetics questionnaire (PIDAQ), and body image concern inventory (BICI). Secondly, it was hypothesized that there would be significant correlations between the BICI and PIDAQ. Thirdly, it was hypothesized that there would be dependencies between a set of predictors (age, gender, and occlusal status), BICI scores, and PIDAQ scores. Finally, it was hypothesized that there would be statistically significant differences among male and female groups with reference to the Body Image Concern Inventory (BICI) and Psychosocial Impact of Dental Aesthetics Questionnaire (PIDAQ).

## 2. Materials and Methods

This was a cross-sectional exploratory study examining the relationships between clinical psychological variables and dental variables as part of normal clinical practice for people affected by malocclusion.

### 2.1. Power and Sample Size Calculation

Assuming an incidence of malocclusion in the general world population of 56% [43], an incidence in the sample under examination (orthodontics clinic) of 70%, considering an alpha significance level of 5%, the minimum number of subjects to be enrolled in order to have a statistical power of 90% is equal to 124 subjects. Therefore, a total sample of 126 subjects was enrolled.

### 2.2. Participants

The sample consisted of 126 subjects aged from 12 to 19 years old (mean: 15.87, SD: 2.35, female: 52.4%, male: 47.6%). The research involved patients from the Orthodontics clinic of the “Gaetano Martino” University Hospital of Messina, Italy, and aimed to explore clinical psychological issues related to dentistry. The evaluations included a clinical evaluation by dentists and psychological tests.

Two orthodontists examined the subjects’ occlusal statuses, evaluating the molar relationship (first, second, and third classes of angle), including patients with moderate crowding of 3–5 mm. The samples had to satisfy all of the following criteria: buccal segment (canines and premolars) eruption was completed; there were no craniofacial anomalies, including cleft lip or palate; all first molars were in place with no proximal caries or restorations; there were no congenital missing teeth or impacted teeth mesial to the first molar; orthodontically untreated. Once the diagnosis of malocclusion was confirmed, the participants completed questionnaires with a licensed psychologist.

Every participant fully completed the questionnaires alone, including information regarding their activities, studies, gender, and age. The complete administration lasted around 30 min for each participant. Before adhering to informed consent or parental consent, participants and parents (if minors) were informed about the anonymous nature of the methods of data processing, as required by the procedures of the ethical committee evidenced by the approval (University of Messina Gaetano Martino University Hospital Ethical Committee approval number: 705; C.E. prot. n. 11–23; date: 6 April 2023).

### 2.3. Statistical Analysis

The data were expressed as means and standard deviations, and the categorical variables as numbers and percentages. The Spearman test was used to evaluate the correlations between variables. The Student’s t-test compared means between gender groups. Multivariate linear regression was used to assess each of the dependencies with the aforementioned set of independent predictors. Statistical analyses were performed using SPSS 26.0 for the Window package. A *p* value smaller than 0.05 was considered to be statistically significant.

### 2.4. Instruments and Variables

#### 2.4.1. Sociodemographic and Medical Variables

Data regarding age, gender, and occlusal status were collected.

#### 2.4.2. Psychological Variables

The following questionnaires were administered individually in the University of Messina Gaetano Martino University Hospital. The model of administration was paper and pencil. All participants fully completed the questionnaire during the scheduled session.

For the evaluation of body image, the Italian Body Image Concern Inventory (I-BICI) was administered [44]. The instrument, designed and validated by Littleton and colleagues [45], aimed to measure the levels of appearance concerns, appearance checking and camouflaging, and social avoidance [46]. As reported by Luca and colleagues, the scale demonstrated excellent sensitivity (96%) and good specificity (67%) for the classification of subjects diagnosed with eating disorders, even with subclinical symptom levels [44]. Exploratory factor analysis supported a two-factor structure.

Items are rated on a 5-point Likert scale (1 = never, 5 = always). The test presents a two-factor structure: the first (dysmorphic symptoms) consists of 12 items referring to feelings of dissatisfaction and shame associated with participants’ appearance; the second (symptom interference) consists of 7 items evaluating the impairment in psychosocial functioning associated with body perception. The Italian validation of the BICI confirmed the internal consistency of the original test, with a Cronbach’s alpha of 0.93. With reference to the two factors, the alpha values corresponded to 0.92 and 0.76, respectively. The use of this test allowed the researchers to exclude subjects with dysmorphophobia and check for a negative body image. The highest scores, ranging from 19 to 95, indicate pathological outcomes. The tool has been successfully used in various clinical fields, as demonstrated by numerous reviews that highlight its role and properties [47,48,49,50].

The Psychosocial Impact of Dental Aesthetics Questionnaire (PIDAQ) (Italian version) [51] is a scale formulated for orthodontic needs assessment and aims to investigate the perception of malocclusion. In consideration of the fact that the orofacial region is an area of prime concern to individuals, the tool aims to capture the subject’s experience through a series of items. Developed and validated by Klages and colleagues in 2005 [52], the instrument has been widely used and translated in several languages [53,54,55,56]. It is a test composed of 28 items and 4 subscales referring to “dental self-esteem” (6 items), “social impact” (8 items), “psychological impact” (5 items), and “aesthetic concern” (3 items). Items are rated on a 5-point Likert scale (1 = not at all, 5 = very much). The internal consistency of the Italian version of the PIDAQ, calculated using Cronbach’s alpha coefficient, varies from 0.79 (aesthetic concerns) to 0.90 (dental self-esteem). The test results make it possible to assess the intensity of these concerns or beliefs, with higher scores corresponding to a greater impact of oral health on quality of life.

## 3. Results

The current study highlighted phenomena and difficulties experienced by subjects affected by malocclusion. The presentation of the results follows the hypotheses stated above. 

Table 1 reports descriptive statistics for numerical variables.

Frequencies for occlusal class were 35.7% for the first class, 34.1% for the second class, and 30.2 for the third class.

Table 2 reports correlational analyses with reference to the first hypothesis.

The first hypothesis concerned the relationships among personal variables such as age and occlusal status, BICI Total Score, dysmorphic symptoms, symptom interference, PIDAQ total score, dental self-confidence, social impact, psychological impact, and aesthetic concerns. This hypothesis concerned correlational analyzes useful for the emergence of significant relationships between variables. As showed in Table 2, with respect to age, two significant and positive correlations were found. PIDAQ—total score and PIDAQ—dental self-confidence were significantly positively associated with age. With reference to occlusal status, several correlations were found to be significant and positive. In these terms, relationships among occlusal status, BICI total score, dysmorphic symptoms, and symptom interference were significant and positive. Concerning occlusal status and PIDAQ, dental self-confidence did not show significant results. Significant and positive correlations were found between occlusal status and PIDAQ—social impact, psychological impact, and aesthetic concerns. No significant correlations were found with reference to BICI variables and age.

Table 3 reports correlational analyses referred to the third hypothesis. 

The second hypothesis concerned correlational analyses between the instruments used. In particular, the variables of the Body Image Concern Inventory (I-BICI) scale were compared to the variables of the Psychosocial Impact of Dental Aesthetics Questionnaire (PIDAQ—Italian version) scale. As reported in Table 3, significant and positive correlations were found among all of BICI—total score and all PIDAQ variables. BICI total score showed positive and significant correlations with PIDAQ total score, social impact, psychological impact, and aesthetic concerns. The significant correlation between BICI total score and dental confidence was negative. Significant and positive correlations were found between BICI dysmorphic symptoms and PIDAQ total score, social impact, psychological impact, and aesthetic concerns. A negative and significant correlation was found between PIDAQ dental self-confidence. Referring to BICY symptom interference, significant and positive correlations were found with PIDAQ total score, social impact, psychological impact, and aesthetic concerns. The correlation between BICI symptom interference and PIDAQ self-confidence was significant and negative. Considering these results, it was possible to attest to the negative and significant correlations referred to PIDAQ dental self-confidence, confirming decreasing levels of self-confidence corresponding to increasing levels of body image difficulties. On the contrary, increased levels of body image difficulties, dysmorphic symptoms, and symptom interference corresponded to higher levels of psychosocial difficulties related to dental pathology, as in the case of social and psychological impact and aesthetic concerns.

Table 4 reports linear regression analyses referred to hypothesis 3. 

The third hypothesis referred to regression analysis, in order to study causal relationships. Multivariate linear regressions were performed in order to evaluate possible dependencies among a set of predictors; specifically, age, gender, and occlusal status, and dependent variables of BICI and PIDAQ questionnaires. Starting from age, we found positive and significant relation that was referred to PIDAQ—dental self-confidence, highlighting how age was a significant predictor of dental self-confidence. A higher number of significant dependencies were found between gender, highlighting its role in the light of BICI—total score, BICI—dysmorphic symptoms, PIDAQ—total score, PIDAQ social impact, and PIDAQ aesthetic concerns. Considering occlusal status, significant dependencies were found with reference to all of BICI’s factors and PIDAQ’s total score, social impact, psychological impact, and aesthetic concerns. All significant dependencies were found were positive.

Table 5 reports differential analyses referred to the fourth hypothesis. 

The fourth hypothesis concerned any statistically significant differences between the group of male and female participants. In this sense, the analyses referred to the variables of the Psychosocial Impact of Dental Aesthetics Questionnaire (PIDAQ—Italian version) and Body Image Concern Inventory (I-BICI) instruments. Most of the variables showed significant differences between groups. Starting from BICI—total score, a significant difference was found with higher mean scores in female group. BICI dysmorphic symptoms showed higher scores in the female group. Statistically significant differences were found with reference to PIDAQ total score, PIDAQ social impact, PIDAQ psychological impact, and aesthetic concerns, with higher scores in the female group.

## 4. Discussion

The present study found significant correlations among dental aesthetics and body image, confirming the importance of clinical psychological dynamics in malocclusion and other medical conditions [25,26,57,58,59]. This is consistent with evidence that there can be psychological consequences deriving from physical conditions [60,61], particularly in chronic conditions [59,62,63,64]. Psychological distress, aesthetics, self-perception, and satisfaction appeared to be prominent in those with dental conditions [65,66,67,68]. Regarding age and occlusal status, recent studies appeared to be in line with the current results [69,70,71,72], suggesting that malocclusion is associated with negative psychological outcomes [73,74,75]. Some studies highlighted the importance of psychological functioning with reference to malocclusion [76,77]. The positive correlations reported in previous studies highlight that self-esteem and self-confidence decrease in the presence of dental difficulties, and can accompany body image difficulties [28,78].

The results pertaining to the first hypothesis appeared consistent with the literature, showing positive correlations between age and occlusal status. These results are consistent with recent studies that showed how malocclusion can affect psychological functioning [74,79,80]. Moreover, implications related to surgical treatments are well known [81,82,83], specifically linked to positive outcomes in psychological and social fields, as well as quality of life and satisfaction.

Relationships between BICI and PIDAQ scales and subscales were all significant. Dysmorphic symptoms, symptom interference, and BICI total score showed positive and significant relationships with all of the PIDAQ dimensions. Consistent with several relevant studies, the relationships between body image and psychosocial dynamics represent key dynamics in malocclusion [75,84,85]. Dysmorphic symptoms as well as related interference were correlated with sociality, as in the case of dental self-confidence, social impact, psychological impact, and aesthetic concerns. These results highlight the impact of physical phenomena in the participants. Similarly, consistent with the literature, dissatisfaction, body image concerns, quality of life, and mental health appeared to represent serious issues among dental patients [86,87,88,89,90]. Considering the psychosocial impact of malocclusion on adolescents and adult subjects, recent studies confirmed the negative role of malocclusion and other dental problems [91,92,93,94,95]. Subsequent psychosocial issues affect subjects’ quality of life and undermine mental health.

The third hypothesis considered the role of some variables on body image and the psychosocial impact of dental appearance, as well as dental issues and related problems. These results suggested that age, gender, and occlusal status play a role. In particular, age appeared to be correlated with self-confidence, highlighting the positive role of aging on confidence and quality of life. This datum appears to be in line with other published articles [69,96], even if some contrary results suggest the need for more attention on the phenomenon [91].

Indeed, the impact of malocclusion has been discussed in some recent studies [97,98,99,100]. In this case, results are both related to dependencies and differences, with higher scores in female subjects for the latter. Considering significant dependencies, all were positive, addressing the role of gender and occlusal status in the involved patients. Age played a direct and predictive role in dental self-confidence, highlighted by a positive and significant correlation. The current and past literature confirm this finding [12,30,34,101,102,103]. Gender as a predictor was significant to BICI’s total score and dysmorphic symptoms. Considering PIDAQ’s dimensions, gender appeared to be a significant predictor for its total score, social impact, psychological impact, and appearance concerns. According to Džemidžić and colleagues [97], psychosocial problems related to malocclusion are common and similar in men and women, with a greater prevalence of appearance concerns in women. In line with these authors, other research studies highlighted the role of gender in the field of malocclusion [104,105,106]. In particular, Wan Hassan et al. [106] suggested that relationships among social impact and dental self-confidence can be influenced by gender. Results showed how predictors can play a relevant role in the onset of psychological maladjustment. Moreover, appearance-related differences have been shown to predict bullying [107,108,109,110,111,112]. Regarding the last predictor, occlusal status, several significant and positive dependencies were found, involving BICI’s total score, dysmorphic symptoms and symptom interference, PIDAQ’s total score, social impact, psychological impact, and aesthetic concerns. Jung [113], in 2010, directly highlighted the effect of this phenomenon on young subjects, showing how malocclusion can affect the self-esteem of young people. Furthermore, Banu and colleagues [114] and Helm et al. [115] analyzed variables such as dissatisfaction, self-perception, and body image and reported a significant impact of dental pathologies. Helm and colleagues [115], through a 15-year follow-up study, suggested how especially occlusal and space anomalies can interfere with body image and self-concept representations along the development phase.

With reference to the significant correlations identified in this study, BICI and PIDAQ variables appeared to maintain different effects between male and female groups. This fourth hypothesis was confirmed through analyses and showed significant differences in the BICI total score and dysmorphic symptoms, PIDAQ total score, social impact, psychological impact, and appearance-related concerns, with greater scores in female participants. Comparing body image results with the contemporary literature, gender differences in body appreciation are known, foreseeing greater efforts and maladjustment in women [116,117,118]. However, this point needs to be further investigated due to a lack of clear evidence.

An important point of discussion concerns the field of interventions. Given the data, it would be necessary to structure interventions useful for reducing maladaptive phenomena. According to a recent study by Crerand et al. [119], interventions to prevent body image disturbances and stigmatization are clinically indicated for both sexes. Some studies have considered interventions aimed at reducing the burden experienced by the patients, as well as intervening in psychosocial terms [120,121]. Despite results linked to psychological interventions, most of the studies are centered on psychological difficulties reduction through orthodontic interventions [26,88,90]. In these terms, it would be important to understand how subjective experience plays a fundamental role for the patient [122]. This entails the need to intervene on an individual level [123]. The conflict dynamics deriving from relational problems pertain to the psychosocial domain, where various interventions have demonstrated their effectiveness.

The data, together with the previously published literature, imply a psychological impact of malocclusion, suggesting the need for a better understanding of the onset and maintenance of clinical psychological difficulties.

## 5. Strengths and Limitations

The present study has several limitations, highlighting the need for further research. The participants were involved in clinical settings, so it would be necessary to extend the number of participants to better reflect the population. Being a preliminary exploratory study, the present contribution only described participants’ variables, preventing the extension of results to other populations. Future studies should include larger observation groups and provide for probabilistic sampling. Moreover, despite the fact that some consistent difficulties clearly emerged through analyses, affective and cognitive dynamics occurring to participants and responsible for the onset of mental sufferance were not included. In these terms, it would be fundamental to better understand dynamics leading subjects’ experience and interfering with self-representation. Further studies should include other fundamental variables in order to deepen understanding of the outcomes and possible psychopathological structures. In conclusion, one of the major domains to take into consideration concerns the field of interventions. As is known in the literature, many of the interventions present refer to dental interventions. Studies referring to pre- and post-intervention evaluations highlight how medical intervention constitutes an opportunity for the reduction of dysmorphic phenomena. In this sense, there is greater attention. The number of studies that highlight the role of psychological interventions in reducing difficulties needs implementation. There emerges a clear need to study in depth the methodologies useful for the treatment of subjects, as well as the data relating to the interventions.

## 6. Conclusions

The present study considered clinical psychological phenomena related to malocclusion in a sample of adolescents and young adults. The results highlighted the impact of dental problems on psychological functioning, self-confidence, body image, and related psychosocial implications. The results highlighted how malocclusion constitutes a serious threat for adolescents’ psychosocial functioning and self-representation constitution. The impact of such physical conditions on participants’ mental health was consistent and clearly represented through significant values. Through the analyses, results appeared to be consistent with past and current trends in the literature, as well as innovative, considering the instruments used. Importantly, aging and worsening of the occlusal status corresponded with higher scores in body image and psychosocial functioning. In light of this, studying directions assumed by body image psychosocial impact of dental condition indexes was useful to understand mutual relationships in line with the significant correlations. Indeed, regression analyses confirmed the predictors’ impact on body image and psychosocial difficulties deriving from malocclusion.

## Figures and Tables

**Table 1 children-10-01691-t001:** Descriptive statistics.

	Mean	Standard Deviation
Age	15.87	2.35
BICI—total score	36.67	17.62
BICI—dysmorphic symptoms	32.08	15.19
BICI—symptom interference	4.58	3.24
PIDAQ—total score	49.41	15.06
PIDAQ—dental self-confidence	14.49	6.71
PIDAQ—social impact	14.84	8.45
PIDAQ—psychological impact	13.28	6.72
PIDAQ—aesthetic concerns	6.88	4.31

**Table 2 children-10-01691-t002:** Correlation analyses among age, occlusal status, BICI, and PIDAQ variables.

	Age	Occlusal Status
BICI—total score	0.047	0.601 **
BICI—dysmorphic symptoms	0.057	0.614 **
BICI—symptom interference	−0.061	0.364 **
PIDAQ—total score	0.176 *	0.357 **
PIDAQ—dental self-confidence	0.222 *	−0.167
PIDAQ—social impact	0.031	0.378 **
PIDAQ—psychological impact	0.001	0.310 **
PIDAQ—aesthetic concerns	−0.079	0.270 **

* *p* < 0.05 (two-tailed); ** *p* < 0.01 (two-tailed).

**Table 3 children-10-01691-t003:** Correlation analyses among BICI and PIDAQ variables.

	BICI—Total Score	BICI—Dysmorphic Symptoms	BICI—Symptom Interference
PIDAQ—total score	0.556 **	0.565 **	0.379 **
PIDAQ—dental self-confidence	−0.259 **	−0.243 **	−0.301 **
PIDAQ—social impact	0.566 **	0.571 **	0.437 **
PIDAQ—psychological impact	0.543 **	0.542 **	0.428 **
PIDAQ—aesthetic concerns	0.663 **	0.417 **	0.371 **

** *p* < 0.01 (two-tailed).

**Table 4 children-10-01691-t004:** Multivariate linear regressions among age, gender, occlusal status (predictors), and BICI and PIDAQ variables (dependent variables).

	Age		Gender		Occlusal Status	
	B (CI)	*p* Value	B (CI)	*p* Value	B (CI)	*p* Value
BICI—total score	0.328 (−0.763/1.418)	0.553	6.795 (1.556/12.035)	0.011 *	11.297 (8.093/14.501)	0.000 *
BICI—dysmorphic symptoms	0.344 (−0.575/1.263)	0.460	6.219 (1.804/10.634)	0.006 *	10.097 (7.398/12.797)	0.000 *
BICI—symptom interference	−0.016 (−0.252/0.219)	0.891	0.576 (−0.554/1.707)	0.315	1.200 (0.508/1.891)	0.001 *
PIDAQ—total score	0.836 (−0.225/1.897)	0.121	6.405 (1.306/11.504)	0.014 *	5.181 (2.063/8.299)	0.001 *
PIDAQ—dental self-confidence	0.656 (0.159/1.153)	0.010 *	−0.310 (−2.697/2.077)	0.798	−1.176 (−2.636/.283)	0.113
PIDAQ—social impact	0.151 (−0.454/.756)	0.621	2.967 (0.061/5.873)	0.045 *	2.977 (1.200/4.754)	0.001 *
PIDAQ—psychological impact	0.093 (−0.395/0.580)	0.708	2.407 (0.058/4.756)	0.045 *	2.077 (0.647/3.507)	0.005 *
PIDAQ—aesthetic concerns	−0.034 (−0.348/0.279)	0.828	1.576 (0.071/3.081)	0.040 *	1.279 (0.359/0.359)	0.007 *

B, beta coefficient; CI, confidence interval. * *p* < 0.05 was considered as significant for the multivariate linear regression analyses.

**Table 5 children-10-01691-t005:** Comparisons between male and female groups.

Variables	Male	Female	*p* Value
BICI—total score	31.216 ± 16.57	41.636 ± 17.18	0.001 *
BICI—dysmorphic symptoms	27.150 ± 14.29	36.575 ± 14.69	0.000 *
BICI—symptom interference	4.066 ± 2.82	5.060 ± 3.53	0.083
PIDAQ—total score	45.416 ± 12.80	53.045 ± 16.10	0.004 *
PIDAQ—dental self-confidence	15.083 ± 6.65	13.954 ± 6.77	0.348
PIDAQ—social impact	12.816 ± 7.24	16.697 ± 9.08	0.009 *
PIDAQ—psychological impact	11.700 ± 5.41	14.753 ± 7.48	0.010 *
PIDAQ—aesthetic concerns	5.816 ± 3.80	7.848 ± 4.55	0.007 *

* *p* < 0.05.

## Data Availability

The datasets generated and analyzed during the current study are original and available on reasonable request from the corresponding author.

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
