# Peer review of "Body Image and Psychological Impact of Dental Appearance in Adolescents with Malocclusion: A Preliminary Exploratory Study"

_children, 2023, doi:10.3390/children10101691_

Round 1
Reviewer 1 Report
I would like to thank the authors for their efforts in studying the OHRQoL in relation to dental aesthetics. The introduction is sufficient and the methodology is detailed, however I believe that the discussion and the conclusion can be enhanced by enriching the content of the discussion, and rewriting the conclusion. The authors describe 4 hypotheses, however only mention two in the discussion part and do not provide clear description of the outcomes after testing these hypotheses.
The conclusion part does not provide the reader with any clear outcomes of the study and can be improved (i.e. refer back to the hypotheses).
Author Response
Dear Reviewer, we thank you for your comments. According to your suggestions, we have performed the required changes.
I'm reporting comments and responsed as follows.
Comment 1:
The introduction is sufficient and the methodology is detailed, however I believe that the discussion and the conclusion can be enhanced by enriching the content of the discussion, and rewriting the conclusion.
Response: Dear Reviewer, we thank you for appreciating introduction and methodology. With reference to discussion and conclusions, we enriched discussions and conclusions. In particular, conclusions have been rewritten.
Comment 2:
The authors describe 4 hypotheses, however only mention two in the discussion part and do not provide clear description of the outcomes after testing these hypotheses.
Response: Thank you very much for your comment. According to your suggestions we enriched the discussion reporting more comparisons and clear descriptions of the outcomes obtained through analyses.
Comment 3:
The conclusion part does not provide the reader with any clear outcomes of the study and can be improved (i.e. refer back to the hypotheses).
Response: Dear Reviewer, thank you for your suggestion. We rewrote conclusions according to your suggestion and referred to hypotheses in order to establish a clear link between hypotheses and results.
Reviewer 2 Report
General Comments:
- The majority of the abstract is unclear with long sentences that make no sense. It must be nearly entirely reworded.
- The English used across the manuscript is poor and seriously disrupts the flow and readability of the paper
- The introduction is poorly worded throughout, uses incorrect English phrases/idioms and misquotes/cites many of the papers referenced.
- The entire paper needs to be strengthened through re-wording. Only then will it be possible to make a call on the quality of the study and the results.
- At present, it is completely unpublishable.
Specific Comments:
Abstract
- “In these terms, physical conditions affecting psychological 12 functioning appear to be a serious cause of maladjustment” – this makes no sense – remove or adjust
- Participants not subjects
- Expand PIDAQ abbreviation
- “A consistent presence of the above-mentioned phenomena emerged, together with significant correlations among age and de class, confirming the positive directions assumed by the involved phenomena.” – this also makes no sense – remove or adjust
- “Significant correlations emerged with reference to BICI and PIDAQ, demonstrating positive directions of all crossings. Significant dependencies emerged 26 highlighting the role of predictors as well as subsequent gender difference, all characterized by higher scores in female group.” – this also makes no sense – improve, remove or adjust
Introduction
- Lines 35-44 – Reword entirely. Poor English, doesn’t flow and certain words are not being used in the correct context e.g. ‘psychic energies’
- Line 67-68: “the detriment of the spontaneity of the oral experience” – what are you even trying to say here? This is really poor.
- Line 72 - Adolescents give more aesthetic value to dental characteristics than males [26,27] – are you missing the word female here?
- Even though you have mentioned BICI and PIDAQ in the abstract, you need to expand in the prose first and then abbreviate also, this must come before any abbreviation, despite the fact you mention it further in the paper in the materials and methods
- All hypotheses are poorly worded and confusing – reword entirely
- No aims or objectives?
Materials and Methods
- ‘Suffering from malocclusion’ is a poor choice of phrase. Reword.
- What are you telling the reader when you write (F: 52.4) – is this a percentage? Ratio? Please clarify.
- Informed consent or assent? Given their age. Parental consent?
- Were the participants known to the orthodontists? Were the orthodontists in the study involved in their clinical care or were they separate? Could this have had any impact?
- ‘among the variables of the following instruments’ – unclear. Name them here or remove this.
- Define occlusal status. And how was this collected during an interview? Surely that needed a clinical exam not an interview?
- How were questionnaires administered? Online? Paper? How long did they take to complete? Could their parents help them fill it out? Did they fill it out in the department or take it home and bring it back in? Who was it handed back in to?
Results
- Where is the breakdown of results from the variables collected in 2.3.1?
- Otherwise these results are so poorly worded in low quality English, I cannot make sense of what they are trying to tell me.
Discussion
- The entire discussion needs to be reworded for clarity. As it stands, the English and choice of words is so poor that it is impossible to interpret. Without clear results, it is impossible to decipher the discussion.
- Where comparisons of the results are made to those in the existing literature, this needs to be expanded more, with more specific examples.
- Strengths and limitations sections?
Conclusions
- The conclusions reached in this study are weak and again, need to be worded better.
Very poor.
Needs completely reworded.
Author Response
Dear Reviewer, thank for your comments. Accorgind to your suggestions we performed the required changes.
I'm reporting comments and responses as follows.
- The majority of the abstract is unclear with long sentences that make no sense. It must be nearly entirely reworded.
Response: Dear Reviewer, according to you comment we revised the abstract making it clearer.
- The English used across the manuscript is poor and seriously disrupts the flow and readability of the paper.
Response: Thank you for your comment. According to the expressed need we revised the whole manuscript involving Nicholas Joseph Lupo, a professional proof reader and English native speaker working at the Mediterranean Journal of Clinical Psychology (https://cab.unime.it/journals/index.php/MJCP/about/editorialTeam)
- The introduction is poorly worded throughout, uses incorrect English phrases/idioms and misquotes/cites many of the papers referenced.
Response: Thank you for your comment. According to your suggestion we revised the introduction removing unnecessary expressions. We performed revisions in order to get clarity.
- The entire paper needs to be strengthened through re-wording. Only then will it be possible to make a call on the quality of the study and the results.
Response: The whole manuscript has been re-worded and revised according to your comments and thanks to the help of the above-mentioned English native speaker/proof reader.
- At present, it is completely unpublishable.
Response: Dear reviewer, we believe that thanks to your comments the paper has gained clarity and a better form. We thank you for your relevant comments contributing to the improvement of our paper.
Specific Comments:
Abstract
- “In these terms, physical conditions affecting psychological 12 functioning appear to be a serious cause of maladjustment” – this makes no sense – remove or adjust
Response: The phrase has been removed and the abstract re-worded.
- Participants not subjects
Response: We replaced subjects with participants.
- Expand PIDAQ abbreviation
Response: Performed change.
- “A consistent presence of the above-mentioned phenomena emerged, together with significant correlations among age and de class, confirming the positive directions assumed by the involved phenomena.” – this also makes no sense – remove or adjust
Response: We removed the phrase and reworded the abstract.
- “Significant correlations emerged with reference to BICI and PIDAQ, demonstrating positive directions of all crossings. Significant dependencies emerged 26 highlighting the role of predictors as well as subsequent gender difference, all characterized by higher scores in female group.” – this also makes no sense – improve, remove or adjust
Response: Fixed.
Introduction
- Lines 35-44 – Reword entirely. Poor English, doesn’t flow and certain words are not being used in the correct context e.g. ‘psychic energies’
Response: We delated the expression and corrected the phrase.
- Line 67-68: “the detriment of the spontaneity of the oral experience” – what are you even trying to say here? This is really poor.
Response: The phrase has been completely reworded.
- Line 72 - Adolescents give more aesthetic value to dental characteristics than males [26,27] – are you missing the word female here?
Response: The word “Female” has been added.
- Even though you have mentioned BICI and PIDAQ in the abstract, you need to expand in the prose first and then abbreviate also, this must come before any abbreviation, despite the fact you mention it further in the paper in the materials and methods
Response: Corrected
- All hypotheses are poorly worded and confusing – reword entirely
Response: Corrected
- No aims or objectives?
Response: We provided for aims and objectives.
Materials and Methods
- ‘Suffering from malocclusion’ is a poor choice of phrase. Reword.
Response: Corrected
- What are you telling the reader when you write (F: 52.4) – is this a percentage? Ratio? Please clarify.
Response: We replaced “F” with “Female”.
- Informed consent or assent? Given their age. Parental consent?
Response: Corrected
- Were the participants known to the orthodontists? Were the orthodontists in the study involved in their clinical care or were they separate? Could this have had any impact?
Response: Clinicians (orthodontists) known the subjects. Clinicians referred to clinical psychology practice met the participants for the first time before the administration of questionnaires. In these terms, we provided for a clinical psychological setting devoid of influences possibly intervening during the administration of questionnaire and foreseen clinical psychological practices.
- ‘among the variables of the following instruments’ – unclear. Name them here or remove this.
Response: Removed.
- Define occlusal status. And how was this collected during an interview? Surely that needed a clinical exam not an interview?
Response: Information has been provided: “Two orthodontists examined the subjects’ occlusal statuses, evaluating the molar relationship (first, second and third classes of Angle) and including patients with moderate crowding of 3-5 mm. The samples had to satisfy all of the following criteria: buccal segment (canines and premolars) eruption was completed; there were no crani-ofacial anomalies, including cleft lip or palate; all first molars were in place with no proximal caries or restorations; there were no congenital missing teeth or impacted teeth mesial to the first molar; orthodontically untreated. Once the diagnosis of mal-occlusion was confirmed, the participants were subjected to questionnaires by a li-censed psychologist.”
“Age, gender and occlusal status were collected and included in the protocol.”
We provided information and reworded the phrase for more clarity.
- How were questionnaires administered? Online? Paper? How long did they take to complete? Could their parents help them fill it out? Did they fill it out in the department or take it home and bring it back in? Who was it handed back in to?
Response: We provided for the following information. “The following questionnaires were administered individually in the University of Messina Gaetano Martino University Hospital. The model of administration was paper and pencil. All participants fully completed the questionnaire during the scheduled session.”
Results
- Where is the breakdown of results from the variables collected in 2.3.1?
Response: For categorial variables, occlusal status was included below Table 1.: “Frequencies for occlusal class were 35.7% for the first class, 34.1% for the second class and 30.2 for the third class.”
We add a row in table one reporting age M and SD.
With reference to gender, descriptive statistics are included in the paragraph “2.1 Participants”.
- Otherwise these results are so poorly worded in low quality English, I cannot make sense of what they are trying to tell me.
Response: we reworded the paragraphs following tables in order to get more clarity.
Discussion
- The entire discussion needs to be reworded for clarity. As it stands, the English and choice of words is so poor that it is impossible to interpret. Without clear results, it is impossible to decipher the discussion.
Response: Dear reviewer, according to your suggestions we reworded the whole paragraph. Moreover, it has been checked during the linguistic revision in order to get clarity.
- Where comparisons of the results are made to those in the existing literature, this needs to be expanded more, with more specific examples.
Response: Thanks for your comment. Consistent comparisons with previous published articles were performed in order to enrich the discussion.
- Strengths and limitations sections?
Response: Following your suggestion we provided for a new paragraph on Strengths and Limitations.
Conclusions
- The conclusions reached in this study are weak and again, need to be worded better.
Response: Thank you very much for the suggestion. The paragraph has been enriched in order to value the study’s results and conclusions.
Reviewer 3 Report
First I'd like to thank authors for sharing their work with us.
The manuscript refers to a cross-sectional study concerning "Body Image and Psychosocial Impact of Dental Aesthetics in Adolescents with Malocclusion". Overall subject is of Children's readers interest. Before considering for publishing authors are encouraged for correcting some issues as:
1. ABSTRACT: as the manuscript refers to a concluded study, it is highly recommended to use past tense, avoiding present tense as LINE 14 - "The present study is aimed at..."
2. KEYWORDS: "Dental issues", "Psychosocial issues" and "Occlusal status" don't match MeSH terms as described. Being a Medical paper, it is highly recommended to correct terms to avoid future searches inconsistencies by scientific community
3. STUDY's LIMITATIONS: not clearly described in text
4: FUTURE STUDIES: authors suggest future studies with larger samples, but what kind of other future studies could be performed to corroborate their findings?
Certificate attachment of proof reading recommended.
Author Response
Dear Reviewer, thank you for your comments. According with your suggestions, we performed the required changes.
I'm reporting comments and responses as follows.
- ABSTRACT: as the manuscript refers to a concluded study, it is highly recommended to use past tense, avoiding present tense as LINE 14 - "The present study is aimed at..."
Response: Dear Reviewer, we thank you for your suggestion. According to your request we corrected the cited text and other errors due to the use of present tense.
- KEYWORDS: "Dental issues", "Psychosocial issues" and "Occlusal status" don't match MeSH terms as described. Being a Medical paper, it is highly recommended to correct terms to avoid future searches inconsistencies by scientific community
Response: Dear reviewer, according to your suggestion we replaced keywords with some new items included in MeSH. Thank you.
- STUDY's LIMITATIONS: not clearly described in text
Response: Dear Reviewer, thank you again for your suggestion. As you can see in the manuscript, we included a new paragraph “6. Strengths and limitations” highlighting the study limits and necessary future directions.
4: FUTURE STUDIES: authors suggest future studies with larger samples, but what kind of other future studies could be performed to corroborate their findings?
Response: Dear Reviewer, within the newly included paragraph and conclusions we included information referred to emerged needs for further studies. Thus, we specified what we believe is needed in terms of sampling, analyses and variables. It would be useful to corroborate our results. Thank you again.
Round 2
Reviewer 2 Report
Comments for Authors:
General Comments:
- Please change ‘relations’ to ‘relationships’ throughout, better English.
- ‘Emerged results’, doesn’t make sense throughout.
- The use of ‘there’, ‘this’ and ‘the’ needs to be reviewed throughout – their use is inconsistent, alongside the misuse of tenses.
- Please review the tenses throughout, especially use of the past tense and use of the word ‘emerged’
- A good effort for a first revision, however, there is still a significant amount of work remaining before this paper is publishable, mainly due to the language used across the paper which makes it difficult to understand for a native English speaker.
- The conclusions section is too long, this should be two small paragraphs at most.
- I would recommend using a different translation/proof reading service, as the majority of the paper is still not in proficient English.
Specific Comments:
Abstract
- You need to be consistent in terms of use of ‘psychosocial’ and ‘psychological’, these are not interchangeable terms
- In addition, I think you’ll need to expand on ‘psycho___ functioning’ i.e. what do you mean when you mention ‘functioning’?
- Expand ‘M’ to ‘mean’ as it’s unclear here and could be interpreted as being shorthand for ‘male’
- Participants were ‘visited’ doesn’t make sense. Reword.
- ‘demonstrating directions assumed by phenomena, in line with the state of the art’ – again, doesn’t make sense, reword.
- Conclusion, you mention ‘psychological’ and ‘psychosocial’, but only mention that ‘psychological’ impact requires more attention.
Introduction
- ‘Referred to bodily representations’ – doesn’t make sense in the context you have placed it in.
- You now write biopsychosocial as ‘bio-psycho-social’ – be consistent across the paper.
- Line 52 needs to have ‘is’ associated with, added.
- Line 64, swallow not swallowing.
- L67 – caries reads better than cavities, this is a scientific paper.
- L72 ‘occlusive’, this is not dental terminology, amend.
- L74 ‘contribute to the onset of psychological defensive patterns’ – this makes no sense, amend.
- L85 ‘decidedly higher’, this is not quantifiable, reword.
- L91-95 – Unclear, reword.
Materials and Methods
- L128 – State ‘mean’ instead of ‘M’
- L162-164 – “A written paper and pencil questionnaire was administered with all participants completing the questionnaire during the scheduled session’. – Reword to this.
Results
- Did you breakdown Class II occlusion into Class II division 1 and Class II division 2? This would provide interesting data.
Discussion
- Again, be careful where you conflate psychological and psychosocial throughout the discussion.
- Line 298 – ‘and’ not ‘ang’
- Line 303 – ‘aesthetic’ not ‘aestetic’
- Unsure what you refer to when you write ‘state of the art’ – please expand and amend.
- ‘Comparing body image results with the state of the art, gender differences in body appreciation are known, foreseeing greater efforts and maladjustment in female subjects’ – this entire sentence doesn’t make sense.
Conclusions
- ‘This data results fundamental in the understand- ing of the participants’ dynamics, so it was clear that aging and worsening of the occlusal status corresponded with higher scores in body image and psychosocial functioning’ – reword.
- ‘how all crossings were significant’ – reword.
- ‘In particular, it was possible to notice how all crossings were significant. This result showed how close these dynamics are in the onset of disturbances. Directly referring to dependencies, analyses permitted to extend directions to causal relations. In line with emerged significant correlations, regression analyses confirmed the predictors’ impact on body image and psychosocial difficulties deriving from malocclusion. Despite it was possible to com- pare results with previously published studies, there is a clear need for further investigations’ – a nice attempt, but this needs entirely reworded for clarity – it doesn’t read well nor make much sense.
- The strengths and limitations section needs to go before the conclusions, at the end of the discussion
o ‘Moreover, despite some consistent difficulties clearly emerged through analyses, affective and cognitive dynamics occurring to participants and responsible for the onset of mental sufferance were not included’ – Needs reworded for clarity, cannot tell what it is currently trying to say’.
o ‘Basal phenomena’ is not a term used in English.
o ‘Psychopathological structures’ – I don’t think you are using this term correctly.
- Reference 82 needs adjusted
Must be improved - English language is currently poor.
Your English language service has not been great - suggest you use another or one suggested by MDPI. If you have already used one suggested by MDPI, use a different one.
Author Response
Dear Reviewer, thank you for your contribution in this article. According to your suggestions, we made the required changes. Moreover, a British native English speaker author in now part of the authors list. In this sense, it was possible to fix all the issues referred to language. We hope that this current version matches your requirements.
General Comments:
- Please change ‘relations’ to ‘relationships’ throughout, better English.
Response: Corrected
- ‘Emerged results’, doesn’t make sense throughout.
Response: Corrected
- The use of ‘there’, ‘this’ and ‘the’ needs to be reviewed throughout – their use is inconsistent, alongside the misuse of tenses.
Response: Corrected
- Please review the tenses throughout, especially use of the past tense and use of the word ‘emerged’
Response: Corrected
- A good effort for a first revision, however, there is still a significant amount of work remaining before this paper is publishable, mainly due to the language used across the paper which makes it difficult to understand for a native English speaker.
Response: Thank you for appreciating the previous revisions. According to our previous response, a British native English speaker is now part of the authors. According to his contribution it was possible to mind more on language and correct all issues.
- The conclusions section is too long, this should be two small paragraphs at most.
Response: Corrected
- I would recommend using a different translation/proof reading service, as the majority of the paper is still not in proficient English.
Response: Corrected
Specific Comments:
Abstract
- You need to be consistent in terms of use of ‘psychosocial’ and ‘psychological’, these are not interchangeable terms
- In addition, I think you’ll need to expand on ‘psycho___ functioning’ i.e. what do you mean when you mention ‘functioning’?
- Expand ‘M’ to ‘mean’ as it’s unclear here and could be interpreted as being shorthand for ‘male’
- Participants were ‘visited’ doesn’t make sense. Reword.
- ‘demonstrating directions assumed by phenomena, in line with the state of the art’ – again, doesn’t make sense, reword.
- Conclusion, you mention ‘psychological’ and ‘psychosocial’, but only mention that ‘psychological’ impact requires more attention.
Response: Dear Reviewer, according to your suggestions, we performed the required changes.
Introduction
- ‘Referred to bodily representations’ – doesn’t make sense in the context you have placed it in.
- You now write biopsychosocial as ‘bio-psycho-social’ – be consistent across the paper.
- Line 52 needs to have ‘is’ associated with, added.
- Line 64, swallow not swallowing.
- L67 – caries reads better than cavities, this is a scientific paper.
- L72 ‘occlusive’, this is not dental terminology, amend.
- L74 ‘contribute to the onset of psychological defensive patterns’ – this makes no sense, amend.
- L85 ‘decidedly higher’, this is not quantifiable, reword.
- L91-95 – Unclear, reword.
Response: Thank you again, we corrected all issues.
Materials and Methods
- L128 – State ‘mean’ instead of ‘M’
- L162-164 – “A written paper and pencil questionnaire was administered with all participants completing the questionnaire during the scheduled session’. – Reword to this.
Response: Corrected
Results
- Did you breakdown Class II occlusion into Class II division 1 and Class II division 2? This would provide interesting data.
Response: Dear Reviewer, thank you for your suggestion. We believe that it would be great to breakdown Class II occlusion into Class II division 1 and Class II division 2. We are working on other populations, so that for us it would be proper to consider this suggestion for further studies. Also, we believe that it would provide interesting data beyond this exploratory study. Thank you again.
Discussion
- Again, be careful where you conflate psychological and psychosocial throughout the discussion.
- Line 298 – ‘and’ not ‘ang’
- Line 303 – ‘aesthetic’ not ‘aestetic’
- Unsure what you refer to when you write ‘state of the art’ – please expand and amend.
- ‘Comparing body image results with the state of the art, gender differences in body appreciation are known, foreseeing greater efforts and maladjustment in female subjects’ – this entire sentence doesn’t make sense.
Response: Corrected
Conclusions
- ‘This data results fundamental in the understand- ing of the participants’ dynamics, so it was clear that aging and worsening of the occlusal status corresponded with higher scores in body image and psychosocial functioning’ – reword.
- ‘how all crossings were significant’ – reword.
- ‘In particular, it was possible to notice how all crossings were significant. This result showed how close these dynamics are in the onset of disturbances. Directly referring to dependencies, analyses permitted to extend directions to causal relations. In line with emerged significant correlations, regression analyses confirmed the predictors’ impact on body image and psychosocial difficulties deriving from malocclusion. Despite it was possible to com- pare results with previously published studies, there is a clear need for further investigations’ – a nice attempt, but this needs entirely reworded for clarity – it doesn’t read well nor make much sense.
- The strengths and limitations section needs to go before the conclusions, at the end of the discussion
o ‘Moreover, despite some consistent difficulties clearly emerged through analyses, affective and cognitive dynamics occurring to participants and responsible for the onset of mental sufferance were not included’ – Needs reworded for clarity, cannot tell what it is currently trying to say’.
o ‘Basal phenomena’ is not a term used in English.
o ‘Psychopathological structures’ – I don’t think you are using this term correctly.
- Reference 82 needs adjusted
Response: Corrected
Finally, we would like to thank the reviewer for the improvement provided. We believe that according to the suggestions, the article reached a better form.
Thank you again and best regards